# Stitching Repair for Delaminated Carbon Fiber/Bismaleimide Composite Laminates

**DOI:** 10.3390/polym14173557

**Published:** 2022-08-29

**Authors:** Jiantao Hua, Suli Xing, Shaohang An, Dingding Chen, Jun Tang

**Affiliations:** College of Aerospace Science and Engineering, National University of Defense Technology, No. 109, Fuyuan Road, Kaifu District, Changsha 410073, China

**Keywords:** carbon fiber/bismaleimide composite, delamination, stitching repair, in-plane compression, bridging effect

## Abstract

Due to the excellent mechanical properties and heat resistance, bismaleimide matrix composite materials have been widely used in aircraft. However, they are susceptible to low-energy impacts, such as bird hits, gravel, tools falling, etc., which can easily result in delamination. The delamination can significantly reduce the compression performance of composites and become a potential hazard for aircraft in service. In this paper, a stitching method developed from the Z-pin manufacturing process was proposed to repair delaminated laminates. Firstly, the delaminated area was stitched by fiber bundles that were pre-impregnated with glue. Then, the fiber bundles threading through the laminate become the pins after the curing process, thus producing the bridging effect between delaminated layers. As a result, the in-plane compressive properties of the laminate are enhanced. The parameters, including the size, number, and position of the stitching hole, for the stitching repair were optimized, and the factors affecting the repair effect were discussed through both finite element analysis and experiments. The results showed that for a carbon fiber/bismaleimide composite plate with a circular delamination roughly 30 mm in diameter, the in-plane compressive strength can be recovered from 54.45% to 84.23% of the pristine plate, and the modulus was fully recovered.

## 1. Introduction

Carbon fiber/bismaleimide (BMI) composites [1,2] possess excellent mechanical properties and heat resistance. They are therefore widely used in aircraft components, such as wing skins, cabin wall panels, etc. Composite structures are susceptible to low-energy impacts throughout production, service, and maintenance. Typical low-energy impacts include tool dropping, runway gravel impact, and bird strikes. These impacts can cause delamination [3,4] inside the components, leading to structural failure under lower than expected loads and a fatal threat to the aircraft. Hence, the repair of delaminated composites is of essential importance.

The primary repair methods for delamination include mechanical connection, scarf repair, and adhesive injection. The thermal expansion coefficients of metal components and composite materials differ significantly for the mechanical linkage, leading to severe internal stresses at the interface in high- or low-temperature environments. In addition, bolts could cause degradation of the aerodynamic characteristics due to the change in the structural surface shape [5]. The study by Mehmet Caliskan [6] showed that the bonded repair was better than the bolted one for single lap joint repairing. The single lap joint [7] and double lap joint cannot improve the delamination resistance in the component, while the bonded scarf repair can solve it, because the bonded scarf repair will remove the delaminated area in the component. However, the bonded scarf repair needs a large working area, because a scarf angle of 3° or even less is required for this method [8,9,10]. Besides, it is challenging to form a curved surface in practice. The adhesive injection can achieve a good effect in the laboratory. However, the injection equipment should be specially designed, and the requirements for resin viscosity and mechanical properties are strict. Otherwise, the saturation of the injection and the repair effect cannot be guaranteed [11]. Cyanate ester resin is reported as a candidate for injection repair due to the low viscosity and high glass transition temperature. Nevertheless, its curing temperature is higher than 177 °C [12], which is unsuitable for in situ aircraft repair [13].

Delamination has varying degrees of impact on the shear strength, fatigue properties, as well as the compressive strength of laminates [14]. Hanhua Li et al. [15] studied the delamination and residual compressive strength of composite components in aircraft. Their research showed that cracks in the matrix and debonding between layers can cause defibration buckling and structural instability. The ultimate load-carrying capacity of the specimen with delamination deficiencies in the skin under compressive load was reduced by 38.58%. The study by A. M. Amaro et al. [16] showed that delamination buckling would occur first when composite laminates with delamination were subjected to compressive loads. Due to the uncoordinated deformation of adjacent layers, the stress at the crack tip increased with the load increasing, making the delamination crack continue to expand until the laminate failed. Therefore, re-bonding of the interlayer could inhibit the propagation of delamination, thus restoring the mechanical properties of the laminates.

Stitching technology is an efficient method to constrain the relative motion of adjacent layers in the manufacturing process. Fiber bundles are used to combine multiple layers into a quasi-3D fabric or to connect independent 2D fabric pieces into a complete structure to enhance the mechanical properties of the composite in the through-thickness direction. After stitching, the composite exhibits excellent interlayer performance, impact tolerance, and compressive strength after the impact [17,18]. Due to its advantages, the stitching of composites has been widely discussed through experiments and numerical simulations [19,20,21,22,23]. Regarding the stitching process, there are many parameters (such as stitch density, thread diameter, stitch spacing, and pitch length) [17,24,25,26] that can affect the performance of composite laminates. Arief Yudhanto et al. [27] demonstrated that the stitch parameters moderately affected the tensile strength of composites. However, stitching reduces the tensile modulus by up to 7% due to in-plane fiber waviness regardless of the stitch parameters. Carbon and Kevlar fibers are recommended as stitching materials due to their high strength, high temperature resistance, and good compatibility with the resin matrix [28].

Based on the discussion above, a stitching repair method developed from stitching technology is proposed for the repair of delaminated carbon fiber/bismaleimide composite laminates. The parameters, including the stitching material, the location, the size, and the number of the stitching holes, are optimized through numerical simulation and experiments, attempting to achieve a high repair efficiency.

## 2. Experimental

### 2.1. Preparation of Delaminated Laminates

A 1500 mm × 1000 mm × 5 mm carbon fiber/bismaleimide (BMI) laminate ([0/45/90/ − 45]${}_{5s}$) was made of ZA50GC-12K/KDC101 prepreg (AVIC Composite Materials Ltd., Beijing, China) through an autoclave (GB150, The Sixth Research Institute of China Aerospace Science and Industry Corporation, Hohhot, China), and the curing process is shown in Figure 1. The specimens were cut into 150 mm × 100 mm for a further compression after impact (CAI) test through a mechanical test machine (WDW-500E, Jinan Time Shijin Testing Machine Co., Ltd., Jinan, China) according to ASTM D 7137-17. Prior to the CAI test, the delamination was generated by a quasi-static indentation test, referring to ASTM D6264-17. The indentation test was performed at a speed of 1 mm/min and stopped at a load of 15 kN, then the specimens were ultrasonically scanned from the surface without indentation. The scanning result is shown in Figure 2. The A scan shows the amplitude of ultrasonic return, which can determine the depth of the delamination. The S scan is the vertical section of the delamination area, which can determine the distribution of delamination in the vertical direction. The C scan can obtain the appearance of the delamination of the specimen at a constant depth, a roughly circular delamination with a diameter of about 30 mm in the specimen, as shown in the C scan. The in-plane compressive properties were tested according to the CAI. The strain tested by the gauges on the two sides of the specimen coincided well, showing that the tests were effective (Figure 3). The compressive strength and modulus of the pristine specimen were 328.7 MPa and 50.73 GPa. The compressive strength and modulus of the delaminated specimen were 179.1 MPa and 24.58 GPa, only 54.49% and 48.97% of the pristine specimens, respectively.

### 2.2. Stitching Repair

In the stitching repair process, fiber bundles stitching the delamination part play an important role in restraining the relative displacements of neighboring layers. The stitching repair process is as follows. First, the delamination area should be recognized. For such impacts, the delamination area is near a circle. Second, a certain number of holes for stitching were drilled and distributed evenly around the edge of the delamination area. Figure 4 shows a case of the delaminated specimen with four holes. In Figure 4, *R* is the diameter of the delamination area, *d* is the diameter of the hole, and *l* is the distance between the centers of the delamination area and the hole. Third, after cleaning the hole carefully, thread fiber bundles impregnated with glue were put through the holes one by one and were repeated until the holes were fulfilled with the fibers. Finally, the stitched fiber bundles were cured for two hours at a temperature of 120 °C in the oven. After curing, the stitching bundles became pins, producing the bridging effect between delaminated layers. Two kinds of fiber bundles, T700 carbon fibers and Kevlar F-12 fibers, were used for stitching in the experiments. The glue used was J-315, which was an epoxy adhesive purchased from the Institute of Petrochemistry Heilongjiang Academy of Sciences.

For comparison, a riveting repair similar to the stitching repair was performed. Instead of stitching, aluminum rivets coated with glue were used to combine the delaminated layers and fill the holes. In this case, the rivet diameter was 2.4 mm and the hole diameter was 2.5 mm. The repaired plates are shown in Figure 5.

## 3. FEM Simulation

In order to guide the experiment, finite element analysis was used to explore and optimize the parameters of stitching repair through the software ABAQUS. A model of carbon fiber/bismaleimide laminate ([0/45/90/−45]${}_{5s}$, 40 layers in total) with a size of 150 mm × 100 mm × 5 mm was established using C3D8R elements and 30 mm-diameter circular delamination in its 4/5, 12/13 and 20/21 layers. Four independent laminates (4 plies, 8 plies, 8 plies, and 20 plies) were constructed as four parts according to the delamination locations. The four parts were assigned material properties, respectively: the circular delaminated area was assigned the material parameters of degraded ZA50GC/BMI, and the rest of the areas were assigned the material parameters of ZA50GC/BMI. Then, they were assembled by setting the “tie” contact property between the surfaces of the plates one by one, except for the delamination area, to compose the delaminated laminate. Besides this, the degraded mechanical properties were attributed to the part adjacent to the delamination area. The normal and degraded mechanical properties are shown in Table 1, and the degradation principle is in reference to [29]. For stitching repair with Kevlar F-12 fibers, unidirectional Kevlar/epoxy (Kevlar/EP) composite rods as the pins were built. The “tie” contact property was set between the rod surface and the hole surface, as shown in Figure 6. The mechanical properties are detailed in Table 1, where E and G refer to the Young’s modulus and shear modulus, and μ is the Poisson’s ratio. Subscripts 1, 2, and 3 correspond to the longitudinal and the two transverse directions, respectively. Boundary conditions are in reference to ASTM 7137-17, and the in-plane compressive load applied was 520 kN.

## 4. Results and Discussion

### 4.1. FEM Results

Twelve cases were simulated to discuss the effects of the number, the location, and the size of the stitching holes, as shown in Table 2. In the cases, Case.*w*/6/2.5 indicates six stitching holes with a diameter of 2.5 mm, and *w* is the ratio of *l*/R. As *w* varies, the positions of the stitching holes change from one case to another. This is to investigate the effect of hole location on the repair of the delaminated laminate. Figure 7 shows the typical stress map of the repaired plate under the in-plane compression of 520 kN. As expected, stress concentrations appear on the crack tip. Table 3 shows the maximum interlaminar shear stress (S${}_{13}$) and through-thickness tensile stress (S${}_{33}$) in Case.*w*/6/2.5. When *w* changed from 0.3 to 0.8, S${}_{13max}$ decreased from 96.34 MPa to 85.04 MPa; however, S${}_{33max}$ increased slightly. When *w* was 1.0, both S${}_{13}$ and S${}_{33}$ increased obviously. Compared with the strength, it is evident that the interlaminar shear stress is much more critical for crack propagation. Therefore, S${}_{13}$ was used to evaluate the repair effect. The nearer the pins are located to the crack tip, the more influential the restraintof the relative displacements of neighboring layers can be, which was proven by the results of S${}_{13}$ in Table 3. Therefore, *w* should be near 1.0. However, when the pins are located at the crack tip, the stress concentrations become severe. Consequently, 0.6 and 0.8 are candidates for *w*.

Figure 8 shows the maximum interlaminar shear stress in Case.0.6/*n*/2.5 and Case.0.8/*n*/2.5; the cases of six stitching holes, i.e., six pins, exhibited better results. On the one hand, the number of pins should be enough to restrain the relative displacements of neighboring layers effectively; on the other hand, stitching holes can be considered as defects, leading to more severe stress concentrations. Consequently, the laminate with six stitching holes reached a good balance in these cases. Furthermore, compared with Case.0.6/6/2.5, Case.0.8/6/2.5 was much better.

Finally, the size effects of the stitching holes, i.e., the diameter of the pins, are discussed through Case.0.8/6/*r*, as shown in Figure 9. The results show that, under interlaminar stress S${}_{13}$, the hole diameter varies from 1.5 mm to 2.5 mm, indicating that pins with only a 1.5 mm diameter are rigid enough. It should be noted that the diameter determines the stiffness of the pins and affects the bonding strength between the pins and the stitching holes. In the FEM model, the surfaces of the pins and the stitching holes are connected directly, ignoring debonding, and the failure of the pins was not considered. Consequently, the conclusion about the size effects of the pins cannot be made from Figure 9 and should be studied further.

### 4.2. Effect of the Size of Stitching Holes

The size effect of stitching holes was studied again through experiments. Figure 10 shows the in-plane compressive strength of delaminated plates after stitching repair with Kevlar F-12 fiber bundles in which six stitching holes and *w* = 0.8 were adopted, and the diameter of the hole was different. The strength of the repaired plates with 2.5 mm-diameter stitching holes was 276.9 MPa, about 84.23% of the pristine plate. The plates repaired with 1.5 mm- and 2.0 mm-diameter stitching holes showed a similar strength, only 67.83% and 67.41% of the pristine one, respectively. Figure 11 shows the in-plane compressive modulus of the plates. For the case of a 2.5 mm diameter, the modulus was 53.5 GPa, about 105.44% of the pristine plate. This may be caused by the additional fiber bundles attaching to the plates, as shown in Figure 5a. For the 1.5 mm and 2.0 mm cases, the modulus was only 43.2 GPa and 39.0 GPa. Obviously, 2.5 mm-diameter holes are recommended.

### 4.3. Effect of the Position and Number of Stitching Holes

Figure 12 shows the in-plane compressive strength of delaminated plates after stitching repair with different *w*, where six stitching holes of a 2.5 mm diameter were adopted. The experiments illustrate a similar result as the FEM research. The nearer the pins are located to the crack tip, the more influential the restraint ofthe relative displacements of neighboring layers and the crack propagation can be, leading to a better repair effect. Figure 13 shows the in-plane compressive modulus. Different stitching locations do not exhibit apparent differences in modulus recovery. The modulus of all the plates recovered to a value higher than 46.0 GPa, about 90% of the pristine plate. Considering the strength recovery effect, *w* = 0.8 is suggested.

Figure 14 shows the in-plane compressive strength of delaminated plates after stitching repair with a different number of stitching holes, where *w* = 0.8 and a 2.5 mm diameter were adopted. As the FEM result, using six stitching holes, i.e., six pins, we achieved a better repair effect. Four pins alone cannot support the sufficient strength and stiffness of the repair, and eight stitching holes result in more defects, reducing the repair effect. Figure 15 shows the in-plane compressive modulus of the plates. Though the strength was not good, the modulus of the plate repaired with eight stitching holes was fully recovered. Another detail that should be noted is that the data discreteness of the case *n* = 4 was much wider. Actually, the tightness of the stitching bundles determines the stiffness and the strength of the cured pins, which are essential for the bridging effect. Because the stitching repair was performed manually, the tightness difference of the bundles in different holes was inevitable. When the number of the pin was small, such as only four pins, the difference resulted in severe structural asymmetry, leading to premature failure and wide discreteness. On the contrary, more pins can weaken the difference, tending to obtain better repair effects.

The experiments illustrate that *w* = 0.8, *d* = 2.5, and *n* = 6 are optimal for the studied delamination destruction.

### 4.4. Effect of Stitching Materials

Figure 16 shows the recovered strength of the plates stitched by Kevlar F-12 bundles, T700 fiber bundles, and aluminum rivets with the parameters *w* = 0.8, *d* = 2.5, and *n* = 6. When stitched with T700 fiber bundles, the strength was 270.6 MPa and 82.34% of the pristine plates. The strength was the lowest repaired with aluminum rivets, only 222.5 MPa and 60.9% of the pristine plates. The tensile modulus of Kevlar F-12 fibers and T700 fibers was about 120 GPa and 210 GPa, respectively. The modulus and strength of T700/EP were also higher than Kevlar F-12/EP. Consequently, the repair effect using T700 fiber bundles should have been better than the repair using Kevlar F-12 bundles. Figure 17 shows the in-plane compressive modulus of the plates stitched by Kevlar F-12 bundles, T700 fiber bundles, and aluminum rivets with the parameters *w* = 0.8, *d* = 2.5, and *n* = 6. When stitched with Kevlar F-12 fiber bundles, the modulus was 53.5 GPa, about 105.44% of the pristine plate. The in-plane compressive modulus improvement was higher than the other stitching materials.

However, the repair effect using the Kevlar F-12 bundle is better; the excellent toughness of Kevlar fibers may cause this. Due to low toughness, the breakage of T700 fibers can be observed in the stitching process, and T700 bundles were broken at the periphery of the stitching holes, as shown in Figure 18a. A similar phenomenon was not observed in the Kevlar repairing plates. Figure 18b shows that until the final fracture, the aluminum rivet had not broken. Consequently, the lousy repair effect of using aluminum rivets can be attributed to the low modulus, only about 70 GPa, and bad bonding strength between the aluminum rivet and the composite surface.

## 5. Conclusions

To sum up, a stitching repair method was proposed for the delamination of composite plates, and the following are the conclusions:

(1) The stitching repair method was proven beneficial for delaminated composite plates. For the ZA50GC/BMI plates with roughly a ϕ30 mm circular delamination, the in-plane compressive strength can be recovered to 84.23%, and the modulus can be fully recovered. Compared with injection repair, stitching repair has the advantages of a simple operation and low requirements for glue. Furthermore, it can obtain better repair effects than the rivet repair.

(2) For the design of the stitching repair, the stitching hole should be close to the edge of the delamination area, and fewer stitching holes are unfavorable to the repair. The bonding strength between the fiber bundle and the stitching hole is essential for sound repair effects. Kevlar fibers comprise a suitable material for sutures compared with carbon fibers.

## Figures and Tables

**Figure 1 polymers-14-03557-f001:**
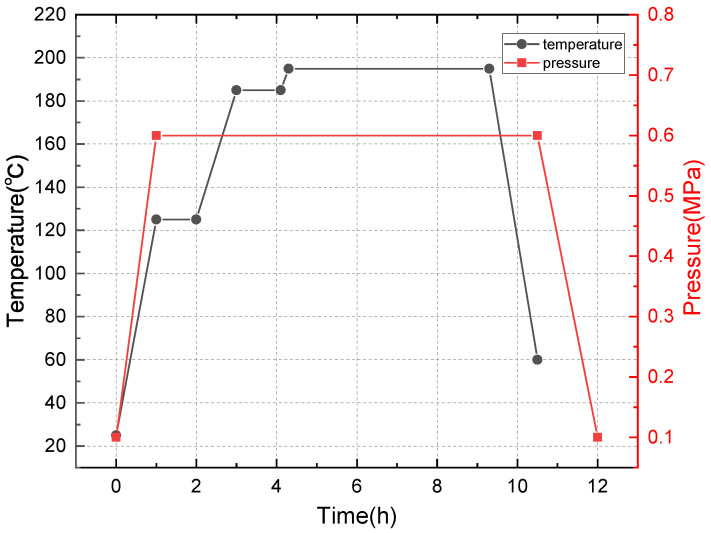
The curing process for the autoclave process.

**Figure 2 polymers-14-03557-f002:**
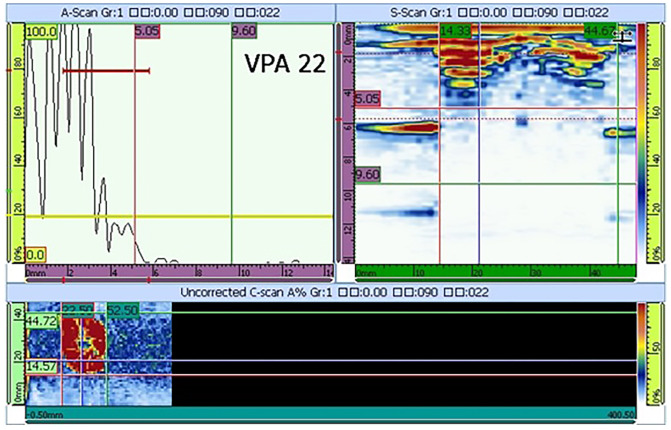
Typical C scan image of the delaminated plate (scan from the back of the indentation).

**Figure 3 polymers-14-03557-f003:**
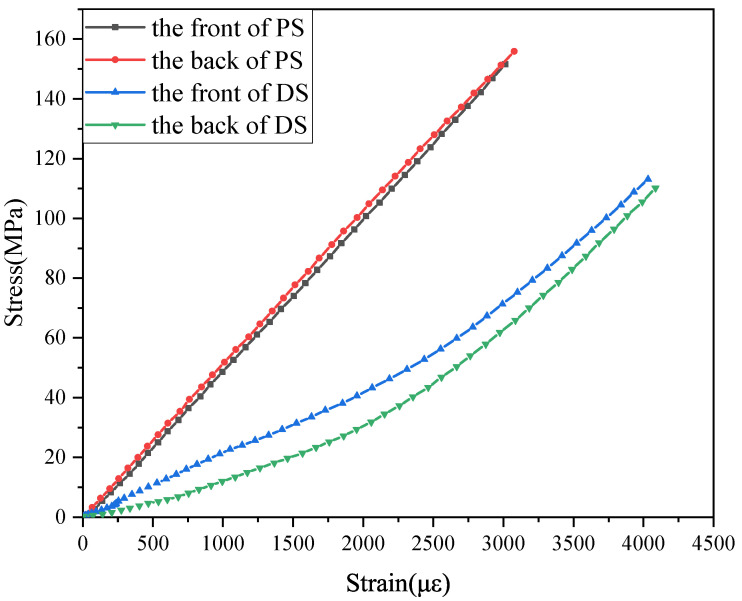
Stress–strain curve of the pristine specimen (PS) and delaminated specimen (DS) under compressive loading (0–3000 με).

**Figure 4 polymers-14-03557-f004:**
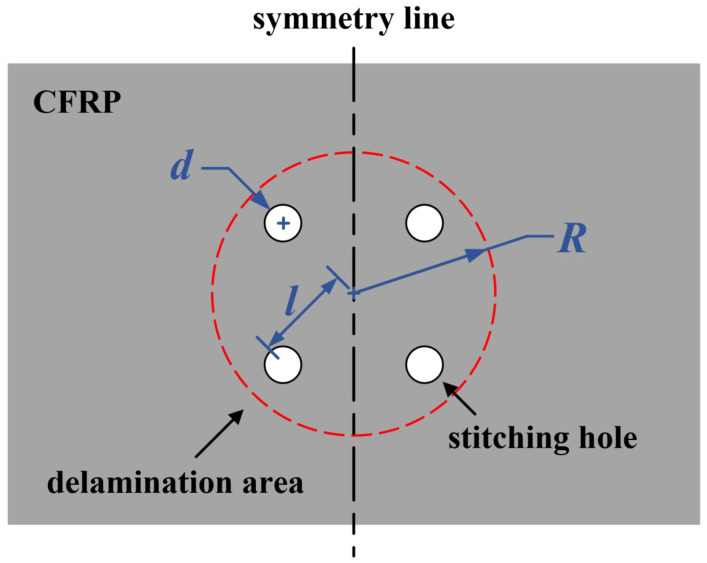
Illustration for the design of repair in which 4 holes are adopted.

**Figure 5 polymers-14-03557-f005:**
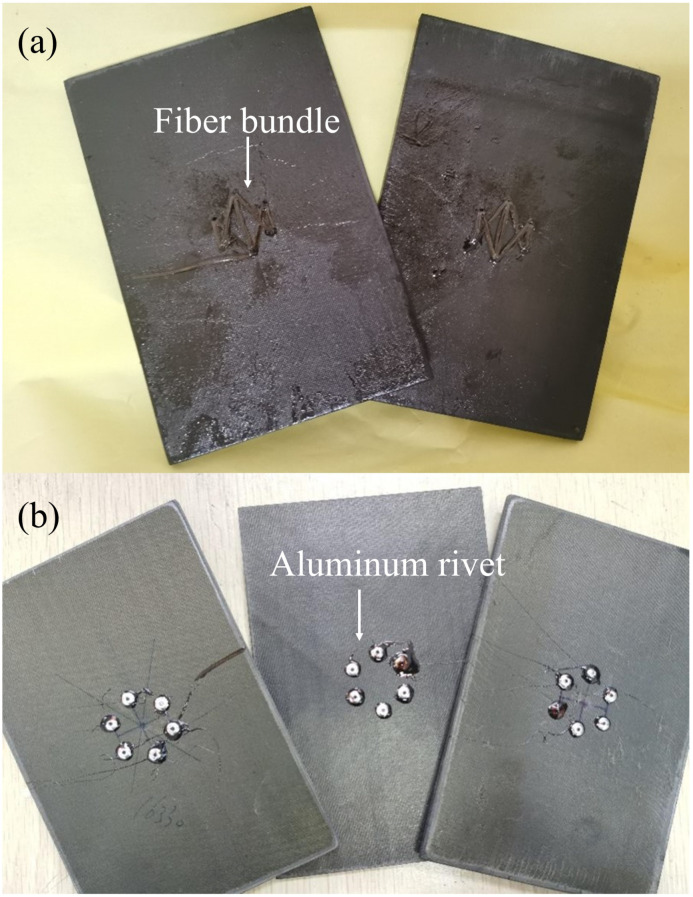
Photograph of plates after (**a**) stitching repair and (**b**) riveting repair.

**Figure 6 polymers-14-03557-f006:**
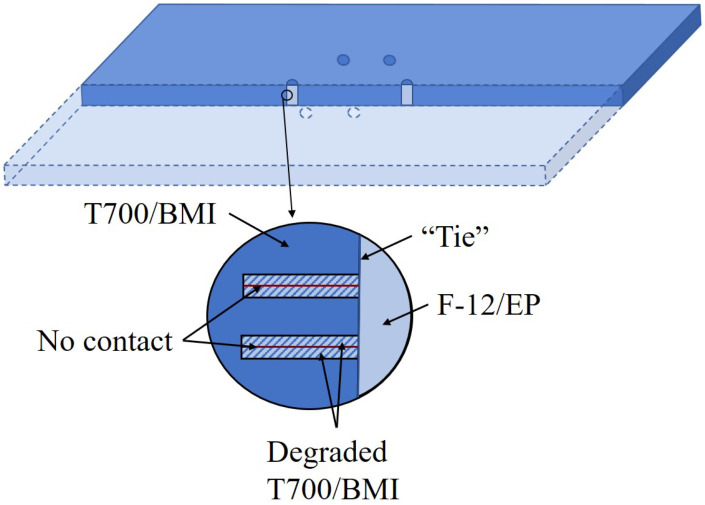
Illustration of the FEM model.

**Figure 7 polymers-14-03557-f007:**
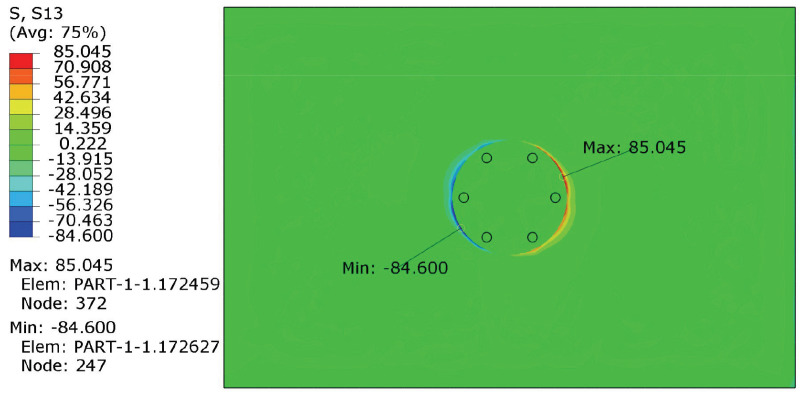
Typical stress distribution of the repaired plate under an in-plane compression of 520 kN.

**Figure 8 polymers-14-03557-f008:**
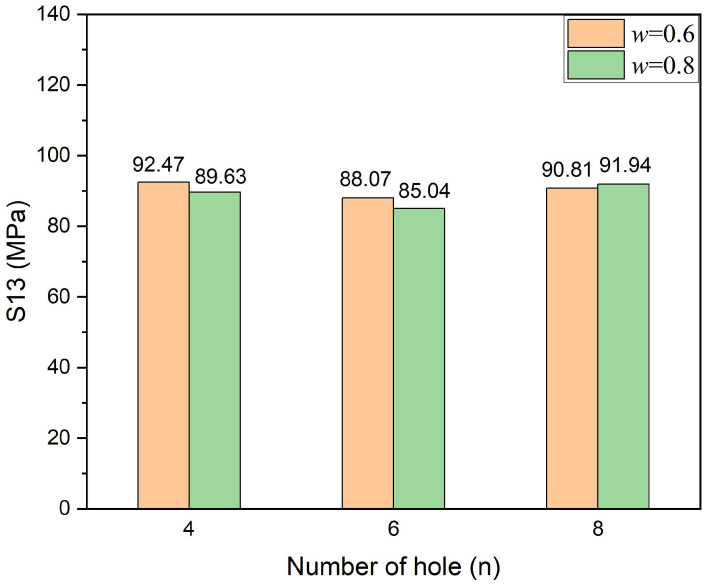
Maximum interlaminar shear stress (S${}_{13}$) in Case.0.6/*n*/2.5 and Case.0.8/*n*/2.5.

**Figure 9 polymers-14-03557-f009:**
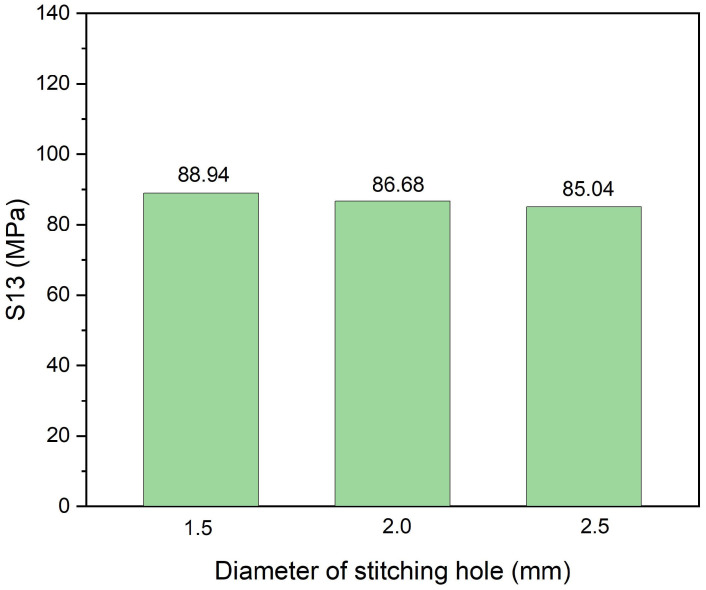
Maximum interlaminar shear stress (S${}_{13}$) in Case.0.8/6/r.

**Figure 10 polymers-14-03557-f010:**
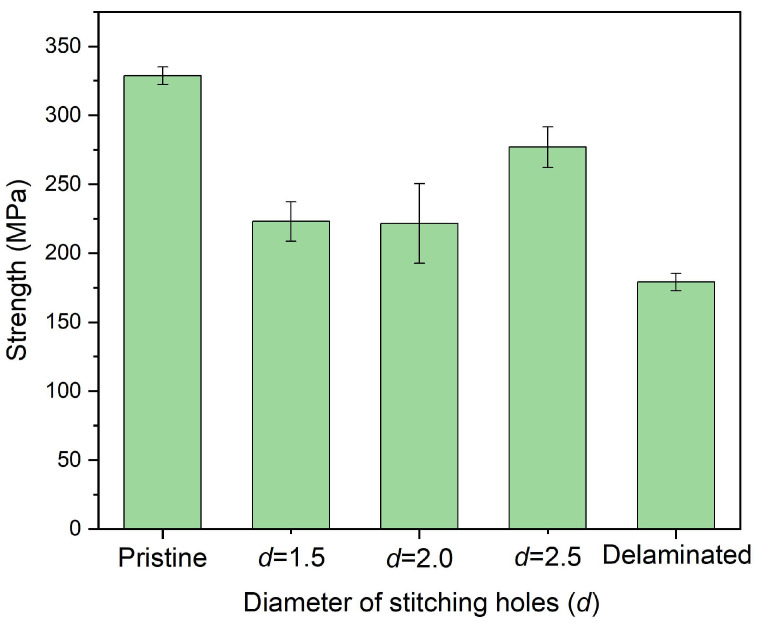
In-plane compressive strength of delaminated plates after stitching repair with different diameters of the stitching holes in which *n* = 6 and *w* = 0.8 were adopted.

**Figure 11 polymers-14-03557-f011:**
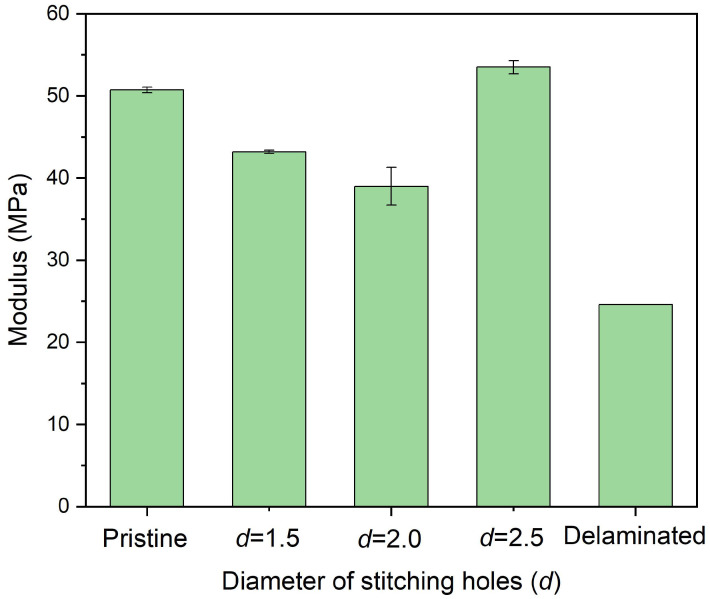
In-plane compressive modulus of delaminated plates after stitching repair with different diameters of stitching holes in which *n* = 6 and *w* = 0.8 were adopted.

**Figure 12 polymers-14-03557-f012:**
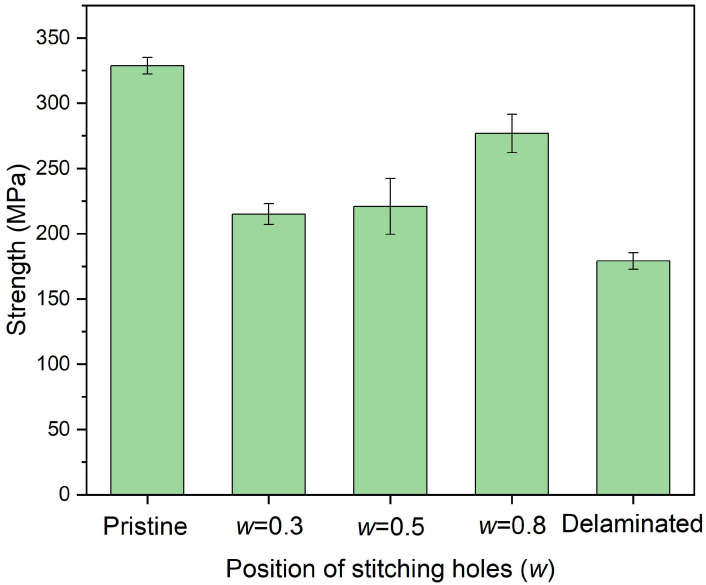
In-plane compressive strength of delaminated plates after stitching repair with different *w* in which 6 stitching holes of a 2.5 mm diameter were adopted.

**Figure 13 polymers-14-03557-f013:**
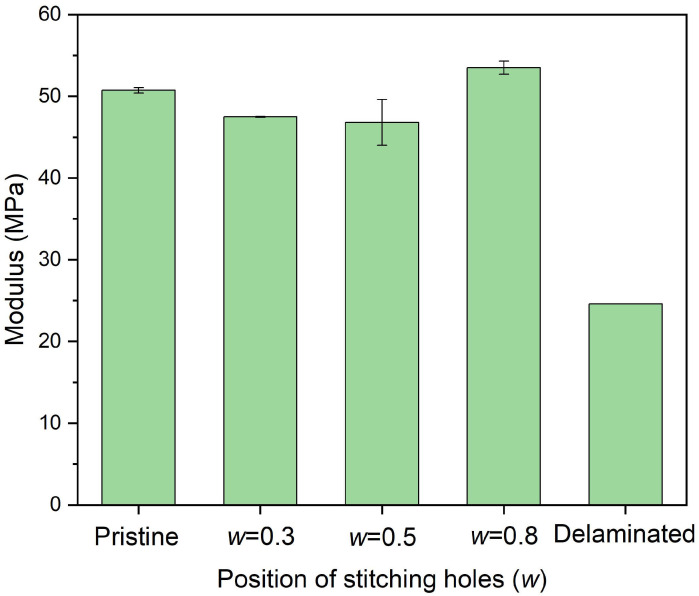
In-plane compressive modulus of delaminated plates after stitching repair with different diameters of stitching holes in which 6 stitching holes of 2.5 mm diameter were adopted.

**Figure 14 polymers-14-03557-f014:**
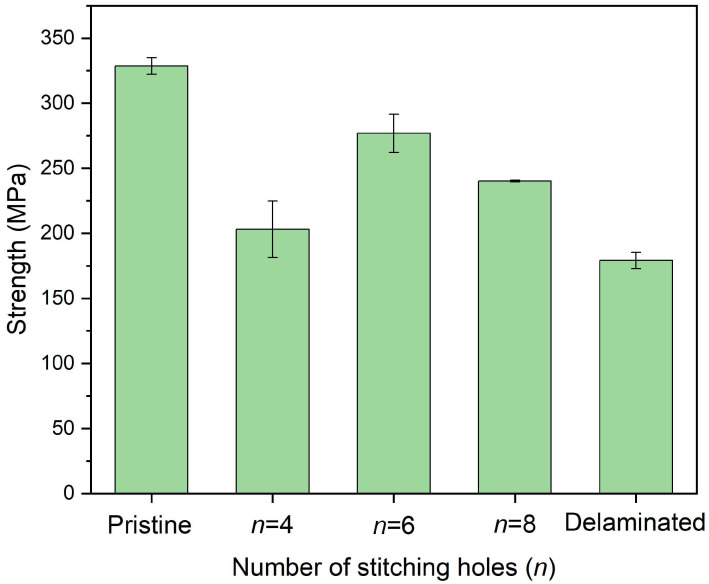
In-plane compressive strength of delaminated plates after stitching repair with different numbers of stitching holes in which *w* = 0.8 and a 2.5 mm diameter were adopted.

**Figure 15 polymers-14-03557-f015:**
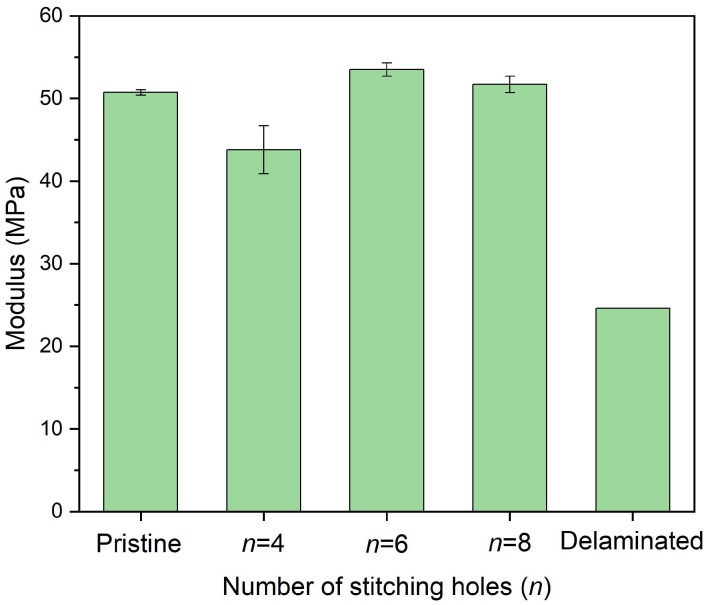
In-plane compressive modulus of delaminated plates after stitching repair with different numbers of stitching holes in which *w* = 0.8 and a 2.5 mm diameter were adopted.

**Figure 16 polymers-14-03557-f016:**
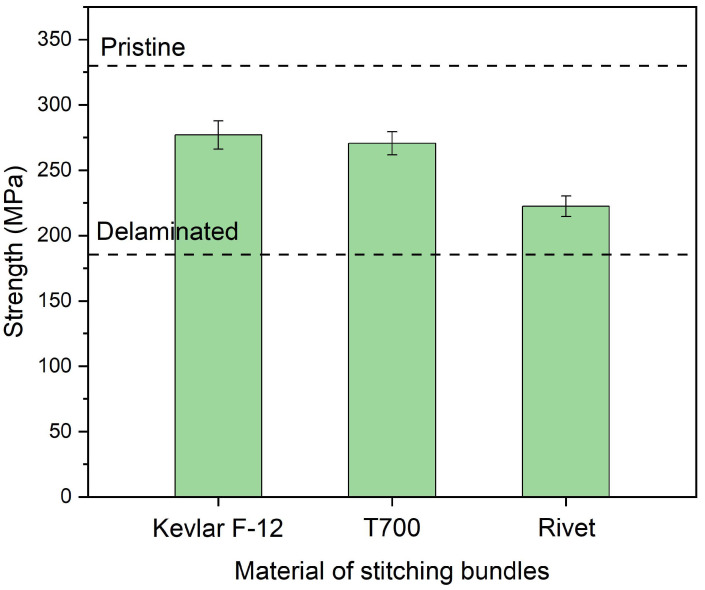
Strength of plates stitched by Kevlar F-12 bundles, T700 fiber bundles, and aluminum rivets with the parameters *w* = 0.8, *d* = 2.5, and *n* = 6.

**Figure 17 polymers-14-03557-f017:**
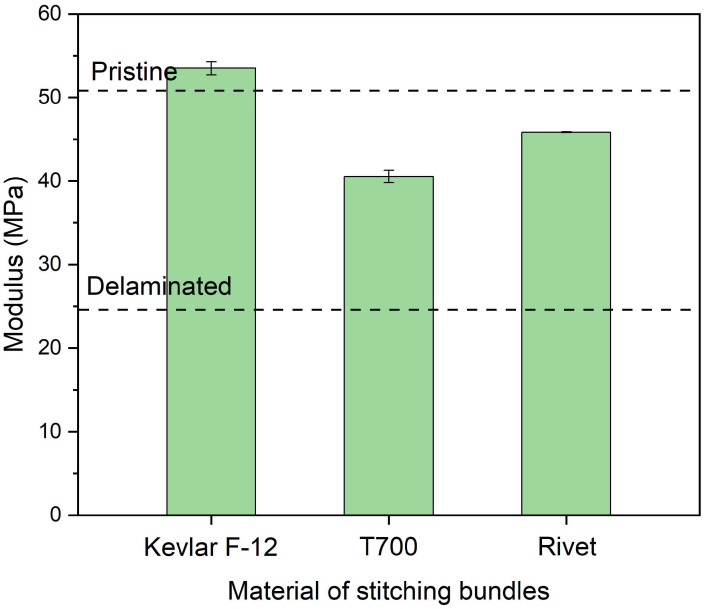
Modulus of plates stitched by Kevlar F-12 bundles, T700 fiber bundles, and aluminum rivets with the parameters *w* = 0.8, *d* = 2.5, and *n* = 6.

**Figure 18 polymers-14-03557-f018:**
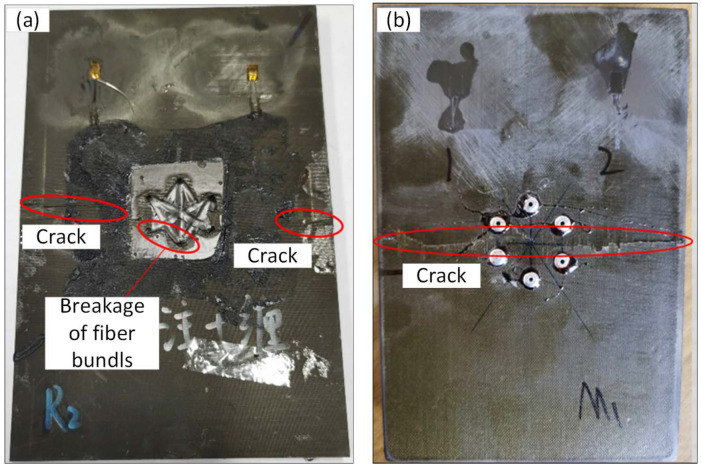
Photographs of broken specimens repaired with T700 fiber bundles (**a**) and aluminum rivets (**b**).

**Table 1 polymers-14-03557-t001:** Engineering constants of unidirectional composite materials for FEM research.

Engineering Constant	ZA50GC/BMI	Degraded ZA50GC/BMI	Kevlar/EP
$E_{1}$/MPa	148,000	20,720	70,000
$E_{2}$/MPa	10,800	1512	4000
$E_{3}$/MPa	10,800	1512	4000
$\mu_{12}$	0.3	0.003	0.4
$\mu_{13}$	0.3	0.042	0.4
$\mu_{23}$	0.4	0.056	0.3
$G_{12}$/MPa	6290	62.9	1700
$G_{13}$/MPa	6290	880.6	1700
$G_{23}$/MPa	3857	539.98	1430

**Table 2 polymers-14-03557-t002:** Research cases in FEM.

Case.*w*/*n*/*d*	*l*/R (*w*)	Number of Holes (*n*)	*d*/mm
Case.0.3/6/2.5	0.3	6	2.5
Case.0.4/6/2.5	0.4	6	2.5
Case.0.5/6/2.5	0.5	6	2.5
Case.0.6/6/2.5	0.6	6	2.5
Case.0.8/6/2.5	0.8	6	2.5
Case.1.0/6/2.5	1.0	6	2.5
Case.0.6/4/2.5	0.6	4	2.5
Case.0.8/4/2.5	0.8	4	2.5
Case.0.6/8/2.5	0.6	8	2.5
Case.0.8/8/2.5	0.8	8	2.5
Case.0.8/6/1.5	0.8	6	1.5
Case.0.8/6/2.0	0.8	6	2.0

**Table 3 polymers-14-03557-t003:** Maximum interlaminar shear stress (S${}_{13}$) and through-thickness tensile stress (S${}_{33}$) in the cases (*n* = 6, *d* = 2.5 mm, different *w*).

	*w* = 0.3	*w* = 0.4	*w* = 0.5	*w* = 0.6	*w* = 0.8	*w* = 1.0	Strength
S${}_{13max}$ /MPa	96.34	94.67	92.03	88.07	85.04	124.00	138
S${}_{33max}$ /MPa	71.38	71.06	71.13	72.37	73.07	80.92	254

## Data Availability

The data presented in this study are available on request from the corresponding author.

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
