# Peer review of "Stitching Repair for Delaminated Carbon Fiber/Bismaleimide Composite Laminates"

_polymers, 2022, doi:10.3390/polym14173557_

Round 1

Reviewer 1 Report

The complex problem of the interlayer connections strength in composites is one of the many current problems of sustainable development. Since this problem is still not completely solved, it is necessary to continue research in label area. The manuscript under review examines the actual problem of joint repair during delamination of composite layered materials made of carbon fiber/bismaleimide..

The manuscript corresponds to the topic of the journal Polymers.

The manuscript contains a sufficient overview, as well as a description of the materials and methods of research, discussion of the results and recommendations for their use (lines 17-75, 76-109, 110-127, 128-220). All structural units of the manuscript, including the abstract and the list of references are logically interconnected and focused on achieving the purpose of the work.

Conclusions confirmed by scientific results and are justified in detail in section 4 (lines 221-233).

The research material is presented clearly; the manuscript is easy to read.

The manuscript meets the criteria of relevance of the topic, novelty and reliability of the results.

The methodology and results of the study are of interest to many potential readers.

However, the following comments appear.

1. Lines 218-220: The authors compare the repair efficiency when using Kevlar and aluminum. The manuscript notes that the poor repair effect when using aluminum rivets can be explained by a low modulus of elasticity. However, it is recommended to pay attention also to the ratio of the final values of deformations of these materials. Perhaps Kevlar adapts better to the deformed state of the layered plate and reduces the stress concentration, including taking into account the differences in temperature deformations of aluminum and composite.

2. Please edit the title of table 1.

3. Table 1, lines 4, 5 and 6: The designation of the Poisson's ratio is not generally accepted. The Poisson's ratio is dimensionless (typo).

4. In Table 1, it is recommended to check the values for Kevlar/EP and other data.

5. It is necessary to edit the header of Table 3 (w/6/2.5.).

6. Useful information on the green background of Figure 6 takes up too little space.

7. The manuscript contains experimental results that may be useful for continuing research in the field of repair and prevention of delamination of composite structures. Therefore, it is recommended to determine a possible direction for further research in this relevant field of applied research.

Author Response

Dear reviewer,

We would like to thank you for your careful reading, helpful comments, and constructive suggestions, which has significantly improved the presentation of our manuscript. Please see the attachment for detailed responses to comments. 

sincerely yours,

Jiantao Hua

Reviewer 2 Report

1.         In line 13, explain the symbol Φ

2.         In section 2.1. Preparation of delaminated laminates, describe in more detail the process of manufacturing samples: the technological mode of autoclave molding, the initial dimensions of the plate, the autoclave brand. What is the thickness of the manufactured plate?

3.         In section 2.1. Preparation of delaminated laminates, specify the impact energy. How many samples were tested for each type of test? Which testing machine was used?

4.         It is recommended to describe the results in more detail in Figure 1. What is depicted in each of the sub figures? How is this data obtained? In which part of the sample was the scan performed?

5.         In section 3 of FEM simulation, describe the FEM simulation procedure in more detail. What program was used? What is the number of finite elements? In table 1, indicate references to literary sources from which data on material properties were taken.

6.         It is recommended in future studies to carry out non-destructive analysis of delaminated plates after stitching repair before conducting mechanical tests. For example, delaminations can be healed by glue leakage during repair.

7.         It is recommended to discuss the possible application of the proposed method in the engineering practice of repairing aircraft structures. How are you planning to continue this work? For example, conducting tensile or shear tests, studying the influence of various climatic conditions, fatigue tests.

Author Response

Dear reviewer,

We would like to thank you for your careful reading, helpful comments, and constructive suggestions, which has significantly improved the presentation of our manuscript. Please see the attachment for detailed responses to comments.

Sincerely yours,

Jiantao Hua

Reviewer 3 Report

This paper reports on the Stitching Repair for Delaminated Carbon Fiber/Bismaleimide Composite Laminates”. The article presents the research topic in an interesting way. Introduction , methodology and reference, results and discussion seems be corrected. Conclusion can be improve.

I have few comments to the manuscript:

1.      All manuscript. Corrected e.g. “[1,2]” to “[1-2]”.

2.      Page 2 line 63. Change from “simulations. [19-23]” to “[simulations [19-23].]”.

Taking into account all comments the manuscript may be published in Polymers after minor revision.

Author Response

(The authors gave the same response as above.)
